# Solar photovoltaic interventions have reduced rural poverty in China

Huiming Zhang[1], Kai Wu[2], Yueming Qiu [3✉], Gabriel Chan[4], Shouyang Wang[5], Dequn Zhou[6] & Xianqiang Ren[1]

Since 2013, China has implemented a large-scale initiative to systematically deploy solar photovoltaic (PV) projects to alleviate poverty in rural areas. To provide new understanding of China's targeted poverty alleviation strategy, we use a panel dataset of 211 pilot counties that received targeted PV investments from 2013 to 2016, and find that the PV poverty alleviation pilot policy increases per-capita disposable income in a county by approximately 7%-8%. The effect of PV investment is positive and significant in the year of policy implementation and the effect is more than twice as high in the subsequent two to three years. The PV poverty alleviation effect is stronger in poorer regions, particularly in Eastern China. Our results are robust to alternative specifications and variable definitions. We propose several policy recommendations to sustain progress in China's efforts to deploy PV for poverty alleviation.

[1] China Institute of Manufacturing Development, Nanjing University of Information Science & Technology, Nanjing 210044, China. [2] School of Finance, Central University of Finance and Economics, Beijing 100081, China. [3] School of Public Policy, University of Maryland, College Park, MD 20742, USA. [4] Humphrey School of Public Affairs, University of Minnesota, Twin Cities, 301 S 19th Ave S, Minneapolis, MN 55414, USA. [5] School of Economics and Management, University of Chinese Academy of Sciences, Beijing 100190, China. [6] School of Economics and Management & Research Centre for Soft Energy Sciences, Nanjing University of Aeronautics and Astronautics, Nanjing 211100, China. ✉email: yqiu16@umd.edu

China's economy has undergone an unprecedented transformation over the past two decades. During this transformation, China has made rapid progress in reducing poverty. In 2000, over 40% of the Chinese population lived below the international poverty line, but by 2010, the poverty rate has decreased to 11.2%. Today, less than 1% of the Chinese population falls below the international poverty line[1]. Since 2013, the Chinese government has identified targeted poverty alleviation as an important national development strategy. This approach has prioritized targeted assistance for the poor, directing resources toward the measurement and alleviation of poverty of individuals, households, and communities[2]. To date, ten initiatives for targeted poverty alleviation have been established, including vocational education and training, providing microcredit, relocation of rural villagers, e-commerce, tourism, planting paper mulberry, entrepreneurial training, and photovoltaics (PV) deployment[3].

The solar energy for poverty alleviation program (SEPAP) in China aims to add over 10 GW of solar capacity to benefit over 2 million citizens by 2020[4]. SEPAP supports solar installations in high-poverty rural villages through three primary types of projects: village-level arrays (for projects generally no more than 300 kW), village-level joint construction arrays (for projects generally no more than 6000 kW), and rooftop installations targeted toward poor villagers (typically several kW).

SEPAP was initially piloted in Hefei and Jinzhai Counties in Anhui Province in 2013. In 2016, the National Development and Reform Commission, the State Council Leading Group for Poverty Alleviation and Development, the National Energy Administration, the China Development Bank, and the Agricultural Development Bank of China jointly issued the Opinions of Photovoltaic Poverty Alleviation Work File, stipulated that by 2020, specifically in the areas both with previous PV poverty alleviation pilot projects and better sunlight conditions, the program should boost overall-village incomes for about 35,000 poverty-stricken villages (for which poverty files have been established) located in 471 counties in 16 provinces. Each of the 2 million poverty-stricken families without capacity to work and for which poverty files have been established (including the handicapped) shall earn an additional income of at least 3000 yuan per household each year from the program. This implies that the sunlight condition is the first order determinant, and the local economic condition is the secondary consideration for the selection of SEPAP poverty alleviation counties. If calculated on the basis of a family of five, the increase in each household's income accounted for more than 10% of the minimum household living standard set by the local government.

By the end of 2018, a total of 15.44 million kW of photovoltaic poverty alleviation has been allocated nationwide, and 2.24 million poor households registered[5]. Several SEPAP projects have achieved notable levels of PV deployment. For example, a 20-megawatt program in Tashkurgan Tajik Autonomous County in Xinjiang, with a total investment of 174 million yuan, provides targeted assistance to 800 poor households[6].

SEPAP projects are structured similarly to community solar programs and other jointly owned renewable energy generation projects developed in the United States, Europe, and other regions. Villagers in such programs and projects appropriate the financial benefits created by a fixed capacity level (share) of an offsite generating facility located in their village. In addition, some revenue of village-level projects can also be withheld for public welfare projects that reduce poverty in the village. For a detailed discussion of SEPAP's historical context and policy development and implementation design, see Geall et al.[4], Li et al.[7], and Murray[8].

Several studies on the intersection of PV deployment and poverty alleviation have focused on the role of PV in providing rural electricity access in locations that do not have access to electric grids or in a few developed countries[9–19]. Moreover, Mandelli et al.[20], Chaurey and Kandpal[21], Rodríguez et al.[22], and Rosas-Flores et al.[23] provide reviews of the deployment of PV for rural electrification.

Only a few prior studies have explored China's experience in rural poverty alleviation through PV deployment and the SEPAP program. Li et al.[24] find that fund shortages are one of the bottlenecks for Chinese PV poverty alleviation projects, which is supported by the analysis of Xu et al.[25] and Wu et al.[26]. Xue[27] and Zhou and Liu[28] emphasize the role of industry structure in eliminating overcapacity and alleviating rural poverty. Geall et al.[4] stress that in the absence of appropriate incentives for local officials and non-state actors, government-led efforts to promote PV-focused energy infrastructure in rural and underdeveloped areas are severely limited. Liao and Fei[29] investigate the SEPAP program in 471 pilot counties in China, with a focus on the information of PV projects and installation capacity instead of county-level data.

These past studies lack a systematic quantitative evaluation expost of the efficacy of the program on its intended goal of reducing rural poverty. To address this research gap, we use propensity score matching and difference-in-difference (PSM-DID) regressions to identify the efficacy of targeted PV poverty alleviation policies on rural disposable income at the county level. We also explore the mechanisms for the effect of PV deployment on poverty alleviation by classifying counties where PV deployment has occurred into two types: counties that also qualify as national poverty-stricken counties and those that are not national poverty-stricken counties. This distinction, which is our second contribution, can disentangle the interrelated effects of poverty alleviation outcomes due to targeted PV deployment from different poverty alleviation programs. We find that the PV poverty alleviation pilot policy increases per capita disposable income in a county by approximately 7–8%. The policy effect generally grows over time two to three years following policy implementation. The PV poverty alleviation effect is stronger in poorer regions.

## Results

**DID model estimation results**. Although SEPAP's intervention covers 471 counties, there are missing data in several variables in the China County Statistical Yearbook, such as rural per capita disposable income of numerous counties, including the pilot counties in Qinghai and Tibet. In this study, we construct a panel of 211 SEPAP pilot counties and a group of control counties from 2013 to 2016. These 211 counties are representative because their GDP accounts for more than 52% of total 471 counties during the sample period. Also, they are distributed in the regions of Eastern, Central, and Western China.

In consideration of the nonrandom assignment of treatment status, special care must be taken to estimate the causal effect of SEPAP on income levels. We implement an approach based on a DID estimator that compares the change in county income before and after SEPAP participation to changes in income over the same time period in control counties. Detailed econometric models and our methodology for constructing our sample can be found in the Methods section.

Table 1 presents the DID estimation results. In Table 1, model (1) controls for industrial structure, the poverty alleviation fund expenditure situation, the land used for agricultural facilities, and education levels. Model (1) finds that the PV poverty alleviation policy is associated with an improved rural disposable income of approximately 7.52%. Model (2) adds a set of additional control variables, which include the degree of marketization across China's regions (a measure consisting of non-state economy, the

**Table 1 The impact of PV policy on natural logarithm of rural per capita disposable income.**

|  | (1) Ln(DISINRURAL) | (2) Ln(DISINRURAL) | (3) Ln(DISINRURAL) |
|---|---|---|---|
| SEPAP | 0.0725*** | 0.0724*** | 0.0446*** |
|  | (9.25) | (9.27) | (4.68) |
| SEPAP*DURATION |  |  | 0.0274*** |
|  |  |  | (5.49) |
| DURATION |  |  | 0.0195*** |
|  |  |  | (5.07) |
| SECONDGDPR | 0.0398 | 0.0091 | 0.0073 |
|  | (1.20) | (0.30) | (0.25) |
| PUBEXINR | −0.0024** | −0.0037*** | −0.0037*** |
|  | (−2.32) | (−3.53) | (−3.38) |
| LN(AGACRE) | 0.0011 | 0.0006 | −0.0003 |
|  | (0.54) | (0.35) | (−0.15) |
| EDUCATION | 0.2559 | 0.1592 | 0.2334 |
|  | (1.10) | (0.72) | (1.07) |
| MKTINDEX |  | 0.0641*** | 0.0662*** |
|  |  | (7.12) | (7.24) |
| LN(SUNHOUR) |  | 0.0659*** | 0.0479*** |
|  |  | (5.31) | (3.78) |
| LN(GDPPROVINCE) |  | −0.0461 | 0.0265 |
|  |  | (−1.36) | (0.69) |
| County FE | Y | Y | Y |
| Year FE | Y | Y | Y |
| Observations | 3598 | 3203 | 3203 |
| Number of Counties | 963 | 857 | 857 |
| Adjusted $R^2$ | 0.01 | 0.06 | 0.08 |

Notes: The dependent variable is the natural logarithm of disposable income of rural people per capita. SEPAP represents whether or not a county was selected for the PV poverty alleviation policy in a specific year. DURATION denotes the exposure time to the program. SECONDGDPR depicts a proportion of the added value of the secondary industry (manufacturing and industrial sector) to GDP. PUBEXINR shows the ratio of public expenditure to revenue. LN(AGACRE) examines the land used for agriculture facilities. EDUCATION estimates the ratio of number of secondary school students to the total population. MKTINDEX represents the marketization index. LN(SUNHOUR) indicates sunlight exposure time. LN(GDPPROVINCE) is the per capita GDP of the province where the county is located. ***, **, and * represent the significance level of 1%, 5% and 10%, respectively. T-statistics are reported in parentheses.

development of product market, the development of factor market, the development of market intermediary organizations, legal environment, and the relation between government and market[30], solar resource (measured as annual solar exposure), and per capita GDP of the province. Model (2) yields a similar policy effect estimate of 7.51%. The issue of the exposure time (to the program) should be addressed since the program had been set up from 2013 to 2016 and the counties start to be covered by the SEPAP in different times. Model (3) adds an interaction item between SEPAP and the exposure time DURATION following King and Jere[31]. In such a situation, the impact of the program is time dependent, since the village collectives might improve their governance of the distribution system and the effect of access to electricity on income can be observed in the long run[32]. The interaction item between SEPAP and the exposure time is significant, with a coefficient of 0.0274.

With regard to control variables in models 1 and 2, the relationship of public finance expenditure with rural disposable income shows a significant negative relationship. The investment in the pilot counties (PUBEXINR) does not have a measurable positive correlation with rural disposable income. The reason may be that the funds generated by PV poverty alleviation projects have not effectively flown to the rural poor due to bureaucratic barriers in income distribution mechanisms that rely on income tax offsets, transfer payments, and direct subsidies. We also find a positive association between the marketization index of a county, solar resource, and rural disposable income.

Supplementary Table 5 in the Supplementary Information shows the results of alternative model specifications. In Supplementary Table 5, model (1) focuses on just the PV poverty alleviation pilots that are also designated as poverty-stricken counties for the treatment group. Other counties with PV pilots are deleted. Model (2) takes all pilot counties of PV poverty

alleviation as the treatment group. Restricting the dataset in this way yields a positive and statistically significant estimate but one that is substantially lower. This model finds that the effect on rural disposable income of PV poverty alleviation policies was 2.6–2.7%. This finding is contrary to hypothesis H3. One possible reason is that, once listed as a national-level poverty-stricken county, a county will be granted additional poverty alleviation funds, discounted loans, and technical support for poverty alleviation projects. These preferential terms may be employed for other poverty alleviation interventions, which could crowd out the benefits of PV for poverty alleviation. Another possible explanation is that further PV investment could exacerbate the rent-seeking behavior of the government. The local government has taken into consideration that if the poverty alleviation policy is implemented successfully in its county and the county is no longer in poverty, it will no longer enjoy the subsidies of the state. We implement further restrictions in Supplementary Table 6 in the Supplementary Information. Model (1) in Supplementary Table 6 excludes PV pilots that are also national-level poverty counties and model (2) excludes all national-level poverty counties. Supplementary Table 6 shows that the poverty alleviation effect of PV policy is significant and large in these counties.

Model (3) in Supplementary Table 5 considers the region-year fixed effects of the three major regions of poverty alleviation interventions, the East, Central, and West. Model (3) uses the full dataset, as the main models in Table 1. This model estimates that the intervention of PV poverty alleviation policy increases annual rural disposable income by 7.51%, which is marginally higher than the estimates in Table 1.

**Test of parallel trend assumption.** A requirement for unbiased DID estimation is the parallel trend assumption. Detailed models

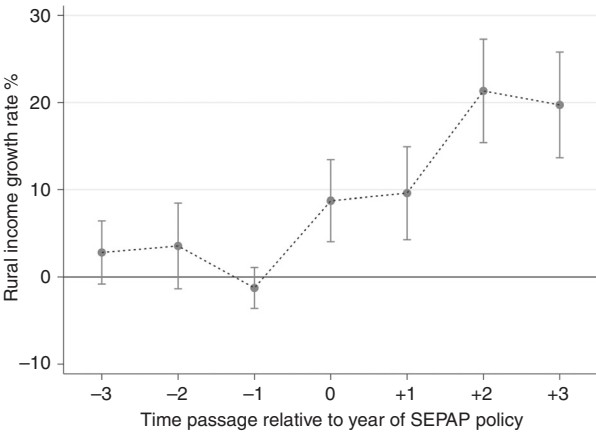

**Fig. 1 Parallel trend using full samples.** Illustration of annual difference in the growth rate of rural income per capita before and after treatment (PV poverty alleviation policy) to demonstrate pre-treatment parallel trend assumption and post-treatment estimated effect. The dots indicate point estimates and the vertical lines indicate 95% confidence intervals. Source data are provided in the Source Data file.

to support this assumption can be found in the Methods section. Supplementary Table 7 in the Supplementary Information shows no significant difference in the trends between the treatment group and the control group in the three years prior to the implementation of SEPAP. Figure 1 displays the results of model (2) in Supplementary Table 7 graphically. The figure and table show that the coefficients for the dummy variables representing the 3 years prior to implementation of SEPAP are not statistically significant at the 5% level regardless of whether control variables are added. Therefore, the parallel trend assumption is satisfied, as the treatment effect before the implementation of PV project is not distinguishable from zero. From the estimates shown in Fig. 1 and Supplementary Table 7, from the year of implementation of SEPAP onward, the impact of PV policies on rural disposable income is positive, generally increasing in magnitude over time, and is statistically significant in each year. The reason for the instantaneous effect of SEPAP is that a number of counties have implemented this program (including figuring out the financing mechanisms and installing PV arrays) for years or many months before the release of the policy. In the 3 years after the implementation of SEPAP, the natural logarithm of per capita disposable income increases, from 9.13% in the year of implementation to 21.80% 3 years after implementation (the estimated effect is slightly higher in the second year after implementation, 23.78%). We interpret estimates of specific effects in post-treatment years with caution, because most of the effect estimated for the third year after implementation is based on the limited number of treatment counties that adopted the PV policy in the early part of our sample period. On average, the number of years observed post-treatment for treatment counties in the panel is 1.77 years.

**DID estimation with a matched sample.** We use PSM to obtain a control group balanced with the treated group, using the same set of county-level characteristics as the matching covariates. Details of the matching approach can be found in the Methods section. We re-estimate the main regression model using the matched sample containing 275 counties. The results in Table 2 are similar to those in Table 1 in direction and magnitude. The balance test in Panel B of Table 2 shows that the $t$ tests of the mean values of the matching variables of the experimental group and control group after the PSM could not reject the null

hypothesis that the treatment group and control group have no significant difference at the 5% significance level. Figure 1 shows parallel trend using full samples and Supplementary Fig. 1 shows parallel trend after matching. Supplementary Fig. 2 illustrates kernel density of treatment and control group.

We conduct several robustness checks, including analyses of counties by income level, region (see Supplementary Note 1), and with different measures of the dependent variable (see Supplementary Note 2). The results in these robustness checks are consistent with those of our main models. As illustrated in Fig. 2, Eastern regions and counties with low per capita GDP appear to experience a greater poverty alleviation effect. Also, to account for the endogeneity bias, we estimate an endogenous treatment effect model in which we use sunlight hours and provincial level per capita GDP to explain the selection of SEPAP (see Supplementary Note 3). The results still show positive impact of SEPAP on rural disposable income.

**Discussion**

The effect of targeted PV poverty alleviation on the natural logarithm of annual rural per capita disposable income is positive with a coefficient of 0.0724, which is statistically significant at the 1% level. The coefficient is decreased to 0.0446 when the interaction term between SEPAP variable and the exposure time is included. The results extend the findings of Geall et al.[4], Zhou and Liu[28], Li[24], and Xu et al.[25] by using quantitative analysis and highlighting the importance of government intervention. Our study is quite different from that of Liao and Fei[29], who focus on the installation capacity instead of income of poverty-stricken families, although we reach the similar conclusion. The policy effect generally grows over time 2–3 years following policy implementation. The poverty alleviation effect of PV policies in eastern regions is slightly greater than in western regions, and poor counties appear to benefit more from PV poverty alleviation policies than wealthy counties. The sunlight exposure time is positively correlated with the annual rural per capita disposable income, consistent with the Bridge et al.[33], which measure the natural conditions and income with annual global solar radiation and consumption per capita, respectively.

The empirical findings in this study suggest the following considerations for policy. First, policy makers need to strengthen the monitoring of implementation in PV counties that are also state-level poverty-stricken counties to counter perverse incentives. The implementation of SEPAP in counties that are also designated as state-level poverty-stricken counties sees gradual increases to rural income. We hypothesize that this is because these counties enjoy access to central poverty alleviation funds and technical support during the implementation of multiple PV and non-PV poverty alleviation policies. This finding raises the possibility that solar PV deployment may act as a substitute for other interventions, perhaps through an income effect. More careful analysis could help identify possible substitution effects. However, in the absence of further analysis, our results suggest that targeting solar PV deployment to counties with few other poverty alleviation interventions has the largest effect on rural incomes. For the state-level poverty-stricken counties, in the process of policy implementation, monitoring must be strengthened, for example by including an improved system of identifying poverty status, tracking the use of poverty-alleviation funds within counties and tracking disaggregated impacts on poverty status of households.

Second, regional differences need to be considered when monitoring the effect of targeted PV poverty alleviation policies. Particular attention needs to be paid to western counties and counties with relatively high per capita GDP, which we estimate

**Table 2 Propensity score matching.**

| | Panel A: matched sample | |
| --- | --- | --- |
| | (1) Ln(DISINRURAL) | (2) Ln(DISINRURAL) |
| SEPAP | 0.0267** | 0.0252** |
| | (2.15) | (2.01) |
| SECONDGDPR | −0.0006 | 0.0156 |
| | (−0.01) | (0.25) |
| PUBEXINR | −0.0023 | −0.0024 |
| | (−1.23) | (−1.29) |
| LN(AGACRE) | 0.0025 | 0.0006 |
| | (0.80) | (0.19) |
| EDUCATION | −0.8796* | −0.3639 |
| | (−1.80) | (−0.71) |
| MKTINDEX | | 0.0865*** |
| | | (4.24) |
| LN(SUNHOUR) | | 0.1370*** |
| | | (4.07) |
| LN(GDPPROVINCE) | | 0.0190 |
| | | (0.24) |
| County FE | Y | Y |
| Year FE | Y | Y |
| Observations | 771 | 771 |
| Number of counties | 275 | 275 |
| Adjusted $R^2$ | 0.02 | 0.08 |

| | Panel B: covariate balance | | | | |
| --- | --- | --- | --- | --- | --- |
| | Sample | Control | Treatment | Diff | T-stats |
| GDPPC | Full | 2.85 | 2.17 | 0.68 | 4.35 |
| | Matched | 2.04 | 2.15 | −0.11 | −0.53 |
| EDUCATION | Full | 0.04 | 0.05 | −0.01 | −3.40 |
| | Matched | 0.05 | 0.05 | −0.00 | −0.49 |
| LNAGACRE | Full | −3.48 | −4.91 | 1.44 | 3.07 |
| | Matched | −4.03 | −4.60 | 0.58 | 1.27 |
| LNSUNHOUR | Full | −1.75 | −1.80 | 0.05 | 0.67 |
| | Matched | −1.80 | −1.82 | 0.03 | 0.28 |

Notes: The dependent variable is the natural logarithm of disposable income of rural people per capita. SEPAP represents whether or not a county was selected for the PV poverty alleviation policy in a specific year. SECONDGDPR depicts a proportion of the added value of the secondary industry to GDP. PUBEXINR shows the ratio of public expenditure to revenue. LN(AGACRE) examines the land used for agriculture facilities. EDUCATION estimates the ratio of number of secondary school students to the total population. MKTINDEX represents marketization index. LN(SUNHOUR) indicates sunlight exposure time. LN(GDPPROVINCE) is used to investigate the per capita GDP of the province where the county is located. ***, **, and * represent the significance levels of 1%, 5%, and 10%, respectively. T-statistics are reported in parentheses.

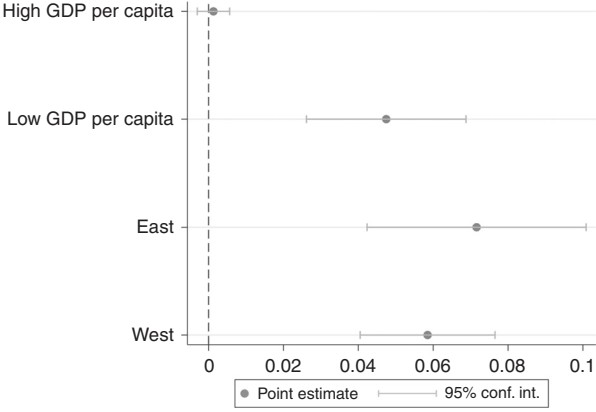

**Fig. 2 Treatment effect estimates by region and economic condition.** Rich counties are those areas with higher income, which tends to correlate with higher marketization levels. Moreover, rich regions are also characterized by effective data and information sharing, as well as cross-sectoral connectivity that extends to farmers and the agricultural sector. The eastern region includes Liaoning, Beijing, Tianjin, Hebei, Shandong, Jiangsu, Shanghai, Zhejiang, Guangdong, Fujian, and Hainan Provinces. The western region includes Xinjiang, Gansu, Qinghai, Inner Mongolia, Ningxia, Shaanxi, Sichuan, Chongqing, Guizhou, Yunnan, and Guangxi Provinces. Source data are provided in the Source Data file.

to have low positive impacts from PV alleviation projects. For future research, investigating the factors that may affect the differences in regional impacts of poverty alleviation programs is important, which could be explained by differences in local political systems, specific uses of poverty alleviation resources, and levels of feed-in-tariffs for different resources. Such further investigation can help better understand the causal mechanisms that drive the poverty alleviation impacts of targeted programs to help design better targeted poverty alleviation policies tailored to regional contexts. The procedures of financial and technical support and policy implementation should reflect the heterogeneity of counties, avoiding one-size-fits-all approaches.

Third, the Chinese experience with PV deployment for rural poverty alleviation provides evidence for other countries considering similar efforts. Other developing regions such as Sri Lanka, Bangladesh, and Palestine have also implemented solar poverty reduction projects[10,16,34]. As deployment of targeted PV interventions for poverty alleviation are expanded, understanding the causal mechanisms and mediating factors that drive desired outcomes to the general application of policy findings is important. Examples of plausible mediating factors include feed-in-tariffs, financing policies, required dispatch of electricity generated by renewable energy resources, and the assistance system for village cadres dispatched by local higher-level governments

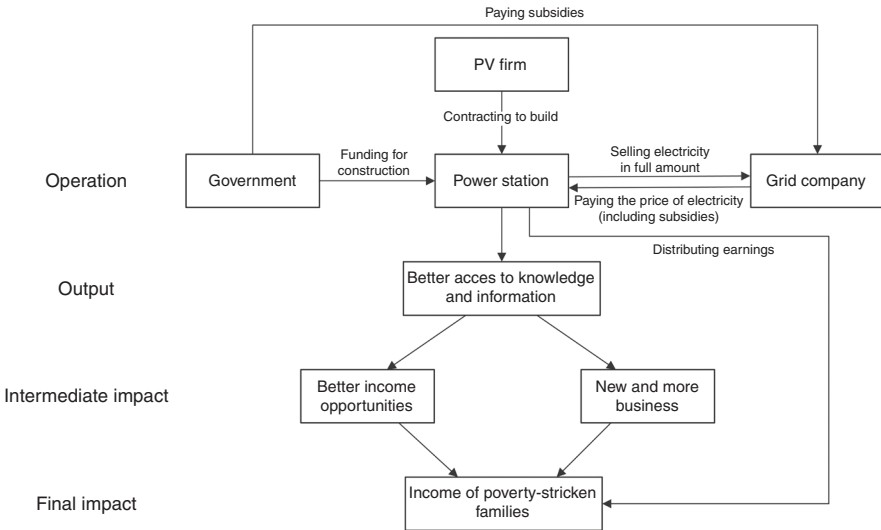

**Fig. 3 Channels through which household income is affected by the SEPAP.** Source: Our own adaptation from Bridge et al.[33], Diallo and Moussa[37], Lenz et al.[38], and Peters and Sievert[39]. The SEPAP program stipulates that the non-residential PV stations funded by governments shall be owned by village collectives. The electricity generated by the power stations is sold to the state grid company after being connected to the grid. The power price sold consists of two parts: desulfurization price and government subsidy. The village collectives shall determine the income distribution mode of the project. Most of the income generated by these projects is directly distributed to eligible poverty-stricken families. The SEPAP also entitles poverty-stricken families to the better access to knowledge and information, and this is an indirect channel affecting income of poverty-stricken families.

(detailed discussion of these factors in Supplementary Table 1). The plausible causal mechanisms of our estimated poverty alleviation effect could be disentangled with future analysis to better understand the extent to which our estimates are driven by direct income effects, high-quality electricity access, employment opportunities in PV, and supporting sectors, or other mechanisms.

Lastly we point out possible directions and challenges for future research. The SEPAP program has only been relatively recently adopted. Therefore, our dataset only covers a short period after the program was implemented. Future analysis should continue monitoring SEPAP and other similar poverty alleviation interventions to better understand whether PV deployment can help address systemic drivers of poverty. This is particularly important given that findings from other contexts where PV technology is deployed in more rural areas of developing countries may not be adapted to local conditions and may face challenges in assuring regular repair and maintenance service[35]. Another challenge is that the cost of the SEPAP program may create political challenges for the continuation of strong financial support from the government. Estimates suggest that the costs of the program could be as high as 30 billion yuan (USD $4.5 billion) over 5 years[36]. Recently, financial support through feed-in-tariffs for general PV projects (not specifically those in SEPAP) has declined, from 0.42 yuan per kWh in 2016 to 0.32 yuan per kWh in 2018. To date, the government has not changed the tariffs for poverty-alleviation PV projects. However, as the costs of the program become more apparent, there may be pressure to reduce financial support, which may increase the difficulty of securing financial support for further projects. Policies should be developed now to enable a smooth transition toward low state support, and further research should develop insights into whether and where continued investment in PV deployment can address poverty alleviation in China.

## Methods

**Econometric model.** PV deployment for poverty alleviation is intended to reduce the burden of energy expenditures by offsetting household energy expenditures in rural communities. The pilot counties selected for PV poverty alleviation

investments may also see growing cumulative benefits over time from discounted interest rates from central and local special poverty alleviation funds, while neither of preferential policies is taking place in the control counties. By jointly developing the capacity needed to develop PV projects with local actors, PV poverty alleviation pilot counties may also see longer-lasting effects through the growth of local businesses and technical capacity for future projects. Under such scenarios, a PV poverty alleviation policy may reduce poverty levels, possibly, with a nonlinear positive effect emerging cumulatively over time as increased disposable incomes lead to reinvestment in rural communities and compounded growth. To identify the channels through which the SEPAP could affect the disposal income, we analyze the following two effects: first, the direct effect due to the income distribution from which households could benefit; second, the indirect effect due to the income effect of benefiting from electrification. As for the SEPAP program established through village-level arrays, the power stations sell generated electricity to the state grid company in full amount, and the later on pay the purchase price which granted with feed-in tariff by the government to the village collectives. Then, as stated in the section of program details, the village collectives distribute most of earnings to the eligible poverty-stricken families in the forms of public welfare posts, small public welfare undertakings, small and micro awards directly. But, there is also an indirect effect (see Fig. 3). The SEPAP program may entitle poverty-stricken families to the better access to knowledge and information, which bring better income opportunities, new business, and income improvement to the beneficiaries.

However, the deployment of PV for poverty alleviation may not guarantee improved rural household incomes in the counties selected for pilot implementation. One concern is that the establishment of pilot counties may result in rent-seeking behavior to benefit from special funds that offset any potential benefits in selected poverty-stricken counties. This implies that the officials in the pilot counties may obtain these funds through the manipulation of the distribution of economic resources, instead of devoting them to poverty alleviation programs. A recent audit on the use of funds in 145 poverty-stricken counties published by People's Daily in 2018 found that violation of discipline and law, loss and waste, and non-standard management amounted to 3.975 billion yuan, which is approximately 6.35% of the total amount of the allocated funds that were audited[40].

Potential factors impede the success of PV deployment for poverty alleviation. First, inadequate data sharing may mean that PV deployment is less effective in targeting poor households and giving benefits for well-off households within selected communities instead. Second, large utility-scale and community-scale PV poverty alleviation projects may fail to distribute benefits or generate revenue for poor households within pilot communities, thus worsening income inequality in rural communities. Third, challenges may be encountered in the formulation and implementation of poverty alleviation policies due to system imperfections, inefficient investment, and unused loans. Finally, a number of PV poverty alleviation projects remain idle or underutilized after completion, and audits conducted to increase accountability may instead be done merely for formality.

In addition to studying the overall average effect of PV poverty alleviation projects, we are also interested in understanding the heterogeneity in influence. PV poverty alleviation pilot counties can be divided into two categories: counties that

also qualify as national poverty-stricken and those that are not national poverty-stricken. Of the 1142 counties we study, 473 are designated as poverty-stricken, of which 175 (37%) were selected for pilot PV poverty alleviation targeting. The remaining 669 counties are not designated as poverty-stricken, of which 36 (5%) were selected for pilot PV poverty alleviation (see Appendix Supplementary Fig. 3).

Photovoltaic poverty alleviation pilot counties refer to areas with good sunshine conditions (annual sunshine time is more than 2000 hours), including some national-level poverty-stricken counties. National poverty-stricken counties, also known as key counties for poverty alleviation and development work, have been identified by the State Council's Leading Group Office of Poverty Alleviation and Development. Once a county is designated as a state-level poverty-stricken county, the central government will allocate major funds to support it and arrange designated assistance. Thus, the funding and technology support for PV poverty alleviation pilot counties that are also designated national-level poverty counties may be higher than in other counties. Therefore, the impact on poverty alleviation may be even greater in these designated counties. This effect is not guaranteed, as the additional funding and support that follows this designation may also incentivize rent-seeking behavior, which in turn weakens the effect of PV deployment on poverty alleviation. On the basis of the analysis of potential positive and negative effects, we propose the following hypotheses.

*H1.* The establishment and implementation of PV poverty alleviation pilot counties has increased rural disposable income.

*H2.* PV poverty alleviation becomes increasingly effective over time (cumulative effect). Therefore, counties that have been pilots for longer periods will see an additional poverty reduction effect.

*H3.* National-level poverty-stricken PV poverty alleviation counties will experience greater effects than non-state-level poverty-stricken counties.

Targeted PV poverty alleviation can be regarded as a policy experiment in which pilot counties are considered treatment counties, and those that do not are considered controls. Owing to the nonrandom assignment of treatment status, special care must be taken to estimate the causal effect of SEPAP on income levels. We implement an approach on the basis of a DID estimator that compares the change in county income before and after SEPAP participation to changes in income over the same time period in control counties. The DID estimator helps to take care of time-invariant unobservable variables, but we admit that it fails to explain the characteristics of time-varying unobservable ones correlating with the treatment. The first round of China's targeted PV poverty alleviation pilot counties was established in 2014. However, 14 counties had already received policy support as early as 2013. Subsequently, the research period is set from 2013 to 2016.

The development levels of different counties in China are highly heterogeneous, and achieving consistent time effects between different counties is difficult. Therefore, before using the DID method, making the experimental group and the control group counties as similar as possible in all aspects is necessary to avoid selection bias. We do this by selecting comparison counties that match the experimental counties in key characteristics. We use the well-established PSM method developed by Heckman[41] and Rosenbaum and Rubin[42]. Although PSM can solve sample selection bias, it cannot solve potential endogeneity arising from omitted variables. DID addresses the endogeneity issue through differencing, but it cannot address the sample selection issue. Therefore, we utilize a combined PSM-DID method, as developed by Heckman et al.[43,44] to identify and evaluate the policy effects of PV poverty alleviation accurately.

Equation (1) shows our basic DID regression

$$\mathrm{Ln(DISINRURAL}_{i,t}) = \alpha_0 + \beta \mathrm{SEPAP}_{i,t} + \lambda_i + \phi_t + \gamma Z_{i,t} + \varepsilon_{i,t}, \qquad (1)$$

where DISINRURAL is the disposable income of rural people per capita, which is our primary measure of poverty levels; $i$ and $t$ are county and year indexes, respectively. $\mathrm{SEPAP}_{i,t}$ is a dummy variable indicating whether a county belongs to the experimental group and equal to 1 if county $i$ has put in place PV poverty alleviation pilot policy in year $t$; otherwise, it takes 0. $\lambda_i$ and $\phi_t$ are country and time fixed effects, respectively. $Z_{i,t}$ is a set of control variables, and $\varepsilon$ is the error term.

The control variables include the proportion of the regional value added GDP by secondary industries, the proportion of public finance expenditure and income, sun exposure, area under cultivation (agriculture), education level, and per capita GDP of the province where the county is located.

The regional industrial structure is measured by the proportion of the value added of secondary industries in regional GDP. A high value indicates a great ability of secondary industries to absorb the employment of rural and urban population, which is conducive to the transfer of rural poor population to the city, reducing the poverty alleviation burden of PV-based poverty alleviation project. Therefore, this variable is expected to improve poverty alleviation effect.

The SEPAP program requires huge investment, and its cost is composed of photovoltaic equipment (including modules, inverters and PV supporting bracket, etc.), engineering procurement construction cost (including civil works, installation, design, access system and project construction management fee, etc.), land use right and capitalized interest, all of which need the funds support. Take one program of GCL new energy holdings limited as a case, the total investment is 179.79 million yuan, among which the equipment investment accounts about 83.39% of the total costs, followed by the engineering procurement construction, 11.69%. The expenditure of poverty alleviation funds is measured by the proportion of public financial expenditure and income. The larger this value is, the

greater the amount of poverty alleviation funds that can be invested. Ceteris paribus, we expect this variable to have a positive impact on poverty alleviation.

Abundant solar resources in a region indicate high PV power generation ability. We expect this variable to have a positive effect on local household income. Both sunlight exposure and average solar radiation are the indicators measuring the abundance of natural conditions. Although the average solar radiation is recognized as one of the determinants for the PV productivity, which has been used by Bridge et al.[33], this indicator is unobservable directly and can only be obtained by conversing from the sunlight exposure time. The conversion is complicated that should incorporate a number of meteorological parameters such as cloud and precipitation, etc. Therefore, we use sunlight exposure time as a proxy for natural conditions.

Regarding land investment, the area under cultivation (agriculture) can be used as an indicator to measure the investment status of PV poverty alleviation. Land under cultivation refers to land with production facilities for crop cultivation or aquaculture. PV poverty alleviation is feasible not only due to solar panels installed on roofs of farmers, barren mountains and deserts, but also on crop cultivation greenhouses or aquaculture fish ponds. More land rent will contribute to large-scale power generation, for example, the village-level plants joint construction arrays will generate more electricity than that of rooftop projects. In theory this indicator is positively correlated with rural per capita disposable income.

Through education, low-income individuals may acquire skills and knowledge that enhance the efficacy of industrial poverty alleviation policies[45]. Education level indicators include number of middle school students, number of primary school students, middle school teachers, and primary school teachers. Compared to the number of primary and secondary school teachers, the proportion of the number of secondary school students in the total population can highlight the local educational output. The improvement of education level helps improve the skills of farmers and is an important measure for poverty alleviation. We recognize the limitations of our education indicator. China implements a 9-year compulsory education for primary and middle school but not for high school. Thus, using the proportion of the number of high school students in the total population to measure local education level may be appropriate. However, the number of middle school and high school students is not listed separately in the China County Statistical Yearbook. Instead, the yearbook reports the total number of secondary school students. For robustness, other indicators such as the proportion of the number of teachers to the total population in the county are also employed, and similar results are reached.

In terms of regional macroeconomic conditions, we use per capita GDP of the province where the county is located. Several indicators, such as economic growth rate, unemployment rate, and inflation rate reflect the regional macroeconomic conditions. In comparison, the per capita GDP of the provinces where the county is located takes regional population into account, and this would be better at highlighting the level of economic development. Better regional macroeconomic performance may have a positive spillover effect on poverty alleviation.

In terms of degree of market development, due to differences in natural endowments and environments, regional economic context and national policies, the degree of marketization varies across China. The ensuing differences in the circulation and distribution of the factors of production in different regions also result in factor market distortions. To measure this distortion we use the market index developed by Wang et al.[30]. The larger the index value, the weaker the factor market distortion. In this case the elements of poverty alleviation can be fully allocated resulting in an improved poverty alleviation effect.

**Data description.** We construct a panel dataset of yearly observations from 2013 to 2016 at the individual county level. The dependent variable (disposable income), the key explanatory variable (PV poverty alleviation policy), and the control variables (industrial structure, poverty alleviation capital expenditure, solar resources, poverty alleviation land resource input level, education level, and regional macroeconomic status) are compiled from the China County Statistical Yearbook. Given that China does not disclose solar irradiance data at the county level, we obtain the data by consulting with the China Meteorological Administration, who collects data from more than 700 meteorological stations across the country. The data is used in the period 2013–2016. As of 2016, China has 1636 counties (excluding districts). After removing the counties with missing data in the statistical yearbook, the sample consists of 1142 counties, including 275 eastern counties, 361 central counties, and 506 western counties.

As shown in Supplementary Fig. 3 in supplementary information, among these 509 counties, 36 pilot counties for PV poverty alleviation are not state-level poverty-stricken. A total of 175 counties are not only pilot counties for PV poverty alleviation but are also state-level poverty-stricken counties, and 298 counties are only state-level poverty-stricken counties. Figure 4 shows the provincial distribution of counties that are selected for SEPAP. The effects of PV poverty alleviation policies studied are short-term, and long-term effects are left for future analysis because of the late occurrence of policies within the panel and the short experimental period.

**Descriptive statistics.** Among the 211 PV counties, 175 are national poverty-stricken counties. Supplementary Table 2 in the Supplementary Information shows that the average per capita GDP is 28,490 yuan. The upper quartile and lower

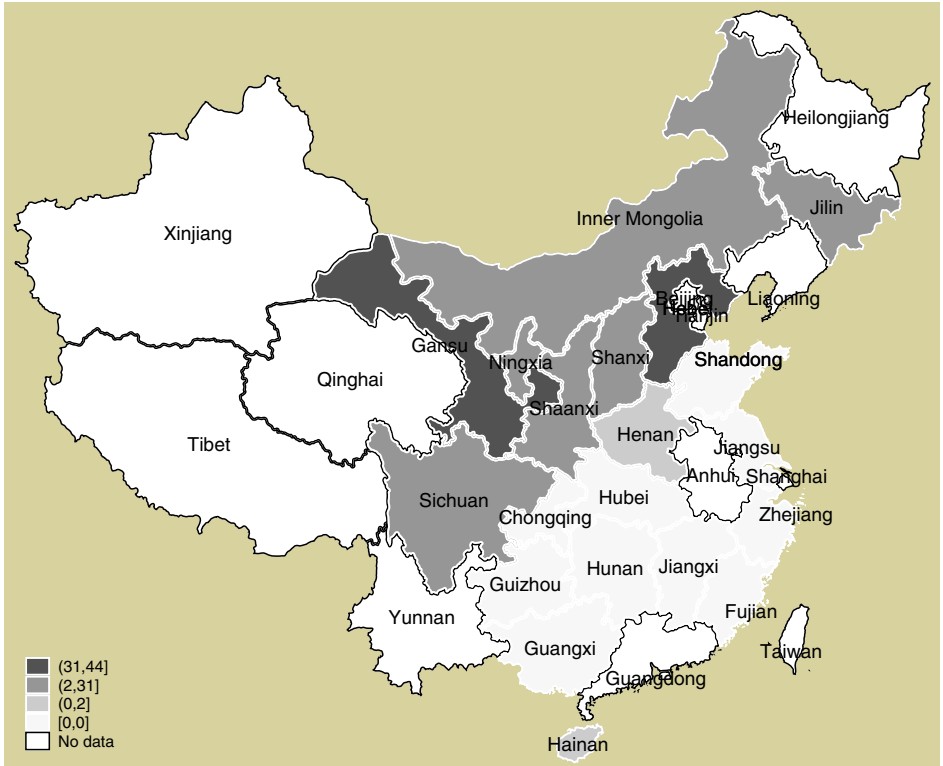

**Fig. 4 Distribution of counties selected for PV poverty alleviation policy in China mainland.** This figure is drafted with 211 sample counties in 2016. The number of photovoltaic counties in each province is calculated. The color depth indicates the size of the number, and the name of the provinces has been marked. There are missing data in some provinces including Qinghai and Tibet, although SEPAP programs have been implemented in these area. Source data are provided in the Source Data file.

quartile are 16,540 yuan and 34,100 yuan, respectively, with a standard deviation of 18,520 yuan. Compared with per capita GDP, the per capita savings balance and the per capita disposable income of rural residents are low, with a mean value of 18,500 yuan and 9140 yuan, respectively. The average per capita disposable income of rural people accounts for about one-third of per capita GDP. The average per capita GDP and median of the provinces in the sample counties are 43,680 yuan and 39,700 yuan, respectively, which are significantly higher than the sample counties. The indicators showed a degree of poverty in the sample counties. The average value of the secondary industry in the sample county is 0.442, and the upper quartile and lower quartile are 0.348 and 0.540, respectively, with a standard deviation is 0.147. The average value of the secondary industry does not exceed 50%, indicating that the contribution of industry to the local economy needs to be improved. The public finance expenditure income ratio is 5.490 and the median is 4.136, indicating a degree of public finance deficit in the sample county. The average number of secondary school students in the sample counties is 0.044, and the upper quartile and lower quartile are 0.036 and 0.051, respectively, which means that the education level of the sample counties needs to be improved. The average marketization index is 6.482, and the variance is 1.328.

**Correlation analysis**. First, the correlation analysis of variables was carried out and the results are in Supplementary Table 3 in the Supplementary Information. The correlation coefficient between the per capita GDP of the province where the sample county is located and the marketization index is 0.72. The correlation coefficient between the marketization index and the per capita disposable income of rural residents is 0.58. The correlation coefficients between other variables are all less than 0.50. These indicate weak correlation between explanatory variables. We made a further analysis of variance inflation factor, and the values were less than 10 (see Supplementary Table 4). Thus the problem of collinearity should not be present.

**Test of parallel trend assumption**. A requirement for unbiased DID estimation result is to satisfy the parallel trend assumption. This requirement means the treatment and control groups should have the same trend before the event occurs, otherwise the DID method will over/under estimate the effect of the event. To test for parallel trends we use the Event-Study method. If the parallel trend hypothesis is established, then the impact of PV policy will only occur after the implementation of the policy (with no significant difference in the trends prior to policy implementation).

To implement the event-study method we estimate Eq. (2)

$$\text{Ln}(\text{DISINRURAL}_{i,t}) = \alpha_0 + \sum_{j=-3}^{3} \beta_j \text{IMPLEMENT}_{i,t-j} + \lambda_i + \phi_t + \gamma Z_{i,t} + \varepsilon_{i,t},$$

(2)

where $\text{IMPLEMENT}_{i,t-j}$ is a dummy variable: when the PV policy is implemented in the pilot county in the year $t - j$ the variable takes 1; otherwise, it takes 0. Therefore, $\beta_0$ measures the current effect of policy implementation in the same year of the implementation; $\beta_{-3}$ to $\beta_{-1}$ measure the effect of 1–3 years prior to the implementation of the policy, $\beta_1$ to $\beta_3$ measure the effect of the policy 1–3 years after the implementation of the policy. The year before the policy implementation is the base year in the model. If $\beta_{-3}$ to $\beta_{-1}$ are statistically significant, then evidence would suggest that the parallel trend hypothesis has violated and that overall results could be driven by selection bias that make the control group an inappropriate counterfactual for the treated group $\beta_0$ to $\beta_4$ measure the dynamic effect of the PV policy over time.

**DID estimation with a matched sample**. First, we use PSM to obtain a comparable control group, and the various county level characteristics are used as the matching criteria. Counties that have not implemented the PV poverty alleviation policy are selected as the control group. The matching variables include the county's per capita GDP, the number of primary and secondary school students accounted for the total population of the county, and the agricultural land area of the county. These matching variables control the macroeconomic development status, education level, and land input of the county. To increase the observation of matching samples, we choose matching ratios of 1:30, 1:40, 1:50, and 1:70, all of which have similar results. The matching ratio of this study is finally set to 1:70.

**Reporting summary**. Further information on research design is available in the Nature Research Reporting Summary linked to this article.

## Data availability

The authors declare that all the data except for the sunlight exposure time supporting the findings of this study are available at [https://github.com/lucy332211/PV-poverty-alleviation-project]. The sunlight exposure time can only be available upon request

because of legal issues. The source data underlying Figs. 1–4 and Supplementary Figs. 1–3 are provided as a Source Data file.

## Code availability

All custom code or mathematical algorithm used to generate results that are reported in this paper and central to its main claims are available at [https://github.com/lucy332211/PV-poverty-alleviation-project].

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

## Acknowledgements

This work was supported by the Major Project of National Social Sciences Fund of China (no. 16ZDA047), National Science Foundation of China (nos. 71503136 and 71834003), and National Social Science Fund (no. 19BGL185). Kai Wu acknowledges financial support from Program for Innovation Research in Central University of Finance and Economics.

## Author contributions

Yueming Qiu, Kai Wu, Huiming Zhang, and Gabriel Chan designed the study, planned the analysis, and edited the language. Xianqiang Ren, Kai Wu, and Huiming Zhang

collected the data. Kai Wu did the data analysis and modified the figure. Yueming Qiu and Gabriel Chan established the models. Huiming Zhang drafted the paper. Shouyang Wang and Dequn Zhou offered the revision suggestions. All authors contributed to the interpretation of findings and approved the final paper.

## Competing interests

The authors declare no competing interests.
