## [Peer Review File · Nature Communications]

Reviewers' comments:

Reviewer #1 (Remarks to the Author):

In this paper, the effects of PV installations targeted for poverty alleviation on rural per-capita disposable incomes in China have been studied. This analysis has been carried out by using mathematical and statistical models using data of 211 pilot counties from 2013 to 2016. Their findings suggest that as a result of large-scale PV implementation, the per-capita disposable income may be increased by approximately 7 %.

In general terms, the paper is well written and the topics included in it are interesting. However, from my point of view several issues should be addressed in order to improve the quality of the manuscript.

1. If there is data available about energy production in the PV installations, new parameters could be included linking this new information with the results of disposable incomes. This is even more important if, as the authors claim, some installations are left unused. The investments needed in each county for alleviating energy poverty and producing the reduction of disposable income could be another potential indicator that should be further explored.

2. For researchers who are not familiar with quantities in Yuan or the cost of living in China, the numbers throughout the manuscript (for example in page 3) might be more comprehensive if represented also as a percentage with respect to the cost of living for a family, as briefly discussed in the descriptive statistics section.

3. Page 4, second paragraph: The information suggests that the electricity produced is never used in the buildings so as to diminish their electricity bills, but instead it is only sold to the grid. Is that true? This might not be optimal from an energetic point of view. Please clarify.

4. Section "Literature review and our contributions", first paragraph. It is true that a relatively small number of studies have examined interventions utilizing solar PV for poverty alleviation. However, many recent studies are missing, so the literature review could be extended with further information. Two examples of this are the studies "Potential energy saving in urban and rural households of Mexico with solar photovoltaic systems using geographical information system" and "Mitigating energy poverty: potential contributions of combining PV and building thermal mass storage in low-income households".

5. Section "Research hypotheses": Please explain for the readers the concept of "rent seeking behavior".

6. Table 1 is a bit difficult for readers to follow. It should be shown in a more comprehensive way.

7. Section "Econometric model": more information about solar availability could be provided, so as to inform the readers about the PV feasibility in the region.

8. Please rephrase the sentence "The effect of PV investment is positive and..." in the abstract.

9. Section "Difference-in-difference (DID) model estimation results": check first sentence.

Reviewer #2 (Remarks to the Author):

Review report on:

"Solar Photovoltaic Interventions Have Reduced Rural Poverty in China"

The paper aims at evaluating the impact of the systematic deployment of solar photovoltaic in rural areas in China. The topic is interesting and need to be carefully addressed. However, the paper needs major revisions in order to proceed further.

In general, the paper is well written. But the program needs to be described in details. Also, the results should be discussed with regard to the existing literature.

Here are some important issues that must discussed.

Major points

The paper lacks a theoretical framework that should establish the theory of change underlining this evaluation. Using this theoretical framework, the paper can identify the channels through which the SEPAP could affect the disposal income. In this program, there are two ways that can be analysed: (i) the direct effect due to the income distribution from which households could benefit, or (ii) the indirect effect due to the income effect of benefiting from electrification.

For the first point, the effectiveness of the income distribution scheme depends on the system of governance established at village level; this issue should be accounted for. For the second issue, the author(s) can build on Diallo & Moussa (2020), Lenz et al. (2017), Peters & Sievert (2016) and Bridge et al. (2016). In these papers, the way access to electricity in general or to solar energy in particular in off grid might affect income are clearly described.

The lack of a theoretical framework led to a misspecification of the evaluation model. In fact, the author(s) state(s) that "This index combines three key factors: proportion of the population in a county in poverty, rural net income per capita...". This implies that the selection of beneficiaries (counties or villages) is endogenous (since it depends on income). Then, the impact assessed is potentially biased.

The issue of the exposure time (to the program) should be addressed since the program had been set up from 2013 to 2016. Including a year fixed effect is not sufficient. An interaction between SEPAP variable and the exposure time is more likely to control for this issue (see King & Jere, 2009). In such a situation, the impact of the program is time dependant; since (i) the village collective might improve their governance of the distribution system and (ii) the impact of access to electricity on income can be observed on long run (Van der Walle et al. 2017).

The choice of control variables needs to be motivated. In particular, the choice of the per capita GDP of the province. Also, the choice of the sunlight exposure time instead of the average solar radiation (that is the more determinant for the PV productivity), see for example Bridge et al. (2016).

Minor points

In Tables 1, 2, S5, S6, S7, S8, S9 and S10, values in parentheses are not standard errors (there are some negative values). Maybe the author(s) mean t-statistics or z-statistics?

Since the SEPAP variable is a binary variable and the main outcome is the log of disposable income, the coefficient β associated with the SEPAP variable should not be interpreted directly. The authors should interpret $100*(e^{\beta} - 1)$ instead; see for example Diallo & Moussa (2020) or Bridge et al. (2016).

References

- Bridge BA, Adhikari D, Fontenla M. Household-level effects of electricity on income. *Energy Econ* 2016; 58: 222e8. <https://doi.org/10.1016/j.eneco.2016.06.008>.
- Lenz L, Munyehirwe A, Peters J, Sievert M. Does large-scale infrastructure investment alleviate poverty? Impacts of Rwanda's electricity access roll-out program. *World Dev* 2017; 89: 88e110. <https://doi.org/10.1016/j.worlddev.2016.08.003>
- Peters J, Sievert M. Impacts of rural electrification revisited e the African context. *J Dev Eff* 2016; 8: 327e45. <https://doi.org/10.1080/19439342.2016.1178320>
- Arouna Diallo, Richard K. Moussa, The effects of solar home system on welfare in off-grid areas: Evidence from Côte d'Ivoire, *Energy* 2020; 194, 116835, <https://doi.org/10.1016/j.energy.2019.116835>.
- Van de Walle D, Ravallion M, Mendiratta V, Koolwal G. Long-term gains from electrification in rural India. *World Bank Econ Rev* 2017; 31: 385e411.

King, Elizabeth M., and Jere R. Behrman. 2009. "Timing and Duration of Exposure in Evaluations of Social Programs." *World Bank Research Observer* 24 (1): 55–82.

Response to reviewers' comments

Reviewer 1#

Comment 1

In this paper, the effects of PV installations targeted for poverty alleviation on rural per-capita disposable incomes in China have been studied. This analysis has been carried out by using mathematical and statistical models using data of 211 pilot counties from 2013 to 2016. Their findings suggest that as a result of large-scale PV implementation, the per-capita disposable income may be increased by approximately 7 %. In general terms, the paper is well written and the topics included in it are interesting. However, from my point of view several issues should be addressed in order to improve the quality of the manuscript.

If there is data available about energy production in the PV installations, new parameters could be included linking this new information with the results of disposable incomes. This is even more important if, as the authors claim, some installations are left unused. The investments needed in each county for alleviating energy poverty and producing the reduction of disposable income could be another potential indicator that should be further explored.

Respond to comment 1

Many thanks for the reviewer's insightful comments. We have tried to search a number of references carefully. In the two official statistical yearbooks in China at the county level, the China County Statistical Yearbook and the China Energy Statistical Yearbook, there is no available data on energy production in the PV installations and the investments needed in each county for alleviating energy poverty. Although it would be much better to choose the two variables suggested by the reviewer, the sunlight exposure time and the ratio of public expenditure to revenue (which are included as control variables in our models) could be used as alternative indirect measures to capture these two aforementioned aspects.

In the future, we expect to conduct a more extensive investigation on the photovoltaic agriculture program under the support of the National Natural Science Foundation.

Comment 2

For researchers who are not familiar with quantities in Yuan or the cost of living in China, the

numbers throughout the manuscript (for example in page 3) might be more comprehensive if represented also as a percentage with respect to the cost of living for a family, as briefly discussed in the descriptive statistics section.

Respond to comment 2

We present the term in percentage regarding the cost of living for a family:

“If calculated on the basis of a family of five, the increase in each household's income accounted for more than 10% of the minimum household living standard set by the local government”.

Comment 3

Page 4, second paragraph: The information suggests that the electricity produced is never used in the buildings so as to diminish their electricity bills, but instead it is only sold to the grid. Is that true? This might not be optimal from an energetic point of view. Please clarify.

Respond to the reviewer 3

In our survey on the PV programs, we find the electricity produced is not used by the households to diminish their electricity bills, but rather sold to the state grid company. This is economically optimal because the electricity generated by SEPAP could be sold to the state grid company at a high price (the grid benchmark price during 2013 to 2015 for the PV poverty alleviation station is 0.9-1.0 yuan/kw•h, and for the year 2016, is 0.8-0.98 yuan /kw•h), while the poverty-stricken family could purchase the electricity from the state grid company at a lower price (the average selling electricity price including tax, for the poverty-stricken family during 2013-2016 was about 0.55 yuan /kw•h).

Comment 4

Section “Literature review and our contributions”, first paragraph. It is true that a relatively small number of studies have examined interventions utilizing solar PV for poverty alleviation. However, many recent studies are missing, so the literature review could be extended with further information.

Two examples of this are the studies "Potential energy saving in urban and rural households of Mexico with solar photovoltaic systems using geographical information system" and “Mitigating energy poverty: potential contributions of combining PV and building thermal mass storage in low-income households”.

Respond to the reviewer 4

Many thanks for the reviewer’s constructive comments. We search a number of studies, and add 14 references which have been published on the high quality journals or newspaper, including the two suggested by the reviewers (see the following references).

[5] Li, Y.S. A photovoltaic ecosystem: improving atmospheric environment and fighting regional poverty. *Technol. Forecast. Soc. Change* 140, 69-79 (2019).

[6] Qin, X., Wu, K.Z. A fight against poverty using photovoltaic power. (*China Business*, March 11, 2019).

[20] Yadav, P., Davies, P.J., Palit, D. Distributed solar photovoltaics landscape in Uttar Pradesh,

- India: Lessons for transition to decentralised rural electrification. *Energy Strateg. Rev.* 26, 1-14(2019).
- [27] Rodríguez, L.R., Ramosb, J.S., Delgado, M. G., Félix, J. L. M., Domínguez, S. Á. Mitigating energy poverty: Potential contributions of combining PV and building thermal mass storage in low-income households. *Energ. Convers. Manage.* 173, 65-80 (2018).
- [28] Rosas-Flores, J.A., Zenon-Olvera, E., Gálvez, D.M. Potential energy saving in urban and rural households of Mexico with solar photovoltaic systems using geographical information system. *Renew.Sust. Energ. Rev.* 116, 1-13(2019).
- [31] Xu, L., Zhang, Q., Shi, X.P. Stakeholders strategies in poverty alleviation and clean energy access: A case study of China's PV poverty alleviation program. *Energ. Policy* 135, 1-13(2019).
- [32] Wu, Y.N., Ke, Y.M., Wang, J., Li, L.W.Y., Lin, X.S. Risk assessment in photovoltaic poverty alleviation projects in China under intuitionistic fuzzy environment. *J Clean. Prod.* 219, 587-600 (2019).
- [39] Diallo, A., Moussa, R.K. The effects of solar home system on welfare in off-grid areas: Evidence from Côte d'Ivoire. *Energ.* 194, 1-12(2020).
- [40] Lenz, L., Munyehirwe, A., Peters, J., Sievert, M. Does large-scale infrastructure investment alleviate poverty? Impacts of Rwanda's electricity access roll-out program. *World Dev.* 89, 88-110(2016).
- [41] Peters, J., Sievert, M. Impacts of rural electrification revisited– the African context. *J. Dev. Effect.* 8(3), 327-345(2016).
- [42] Bridge, B.A., Adhikari, D., Fontenla, M. Household-level effects of electricity on income. *Energy Econ.* 58, 222-228(2016).
- [45] King, E.M., Jere, R. B. Timing and duration of exposure in evaluations of social programs. *World Bank Res. Obser.* 24 (1), 55–82(2009).
- [46] Van de Walle D., Ravallion, M., Mendiratta, V., Koolwal, G. Long-term gains from electrification in rural India. *World Bank Econ. Rev.* 31(2), 385-411(2017).
- [54] Ge, J.P., Lei, Y.L. Mining development, income growth and poverty alleviation: A multiplier decomposition technique applied to China. *Resour. Policy* 38, 278-287(2013).

Comment 5

Section “Research hypotheses”: Please explain for the readers the concept of "rent seeking behavior".

Respond to the reviewer 5

The concept of rent-seeking was introduced in 1967 and popularized in 1974 by the economist, Anne Krueger. In this study, we add more interpretation combining the PV poverty alleviation programs:

“This implies that the officials in the pilot counties may obtain these funds through the manipulation of the distribution of economic resources, instead of devoting them to poverty alleviation programs.”

Comment 6

Table 1 is a bit difficult for readers to follow. It should be shown in a more comprehensive way.

Respond to the comment 6

We make several changes in the Table 1. We add ‘Ln(DISINRURAL)’ to denote the dependent variable in each column. The same thing is done for Table 2, and Table S5 to Table S7. Also, we change the format of these tables. We hope that the revisions could meet reviewer’s requirements.

Comment 7

Section “Econometric model”: more information about solar availability could be provided, so as to inform the readers about the PV feasibility in the region.

Respond to the comment 7

In the revised manuscript, more information about solar availability is provided:

“Photovoltaic poverty alleviation pilot counties refer to areas with good sunshine conditions (annual sunshine time is more than 2000 hours)”.

We also add the following statement about the data source for solar irradiance data:

“Given that China does not disclose solar irradiance data at the county level, we obtain the data by consulting with the China Meteorological Administration, who collects data from more than 700 meteorological stations across the country”.

Comment 8

Please rephrase the sentence “The effect of PV investment is positive and...” in the abstract.

Respond to the comment 8

Special thanks for the reviewer. The original sentence is “The effect of PV investment is positive and significant within the year of policy implementation and is more than twice as high two to three years subsequently”.

The revised sentence is:

“The effect of PV investment is positive and significant in the year of policy implementation and the effect is more than twice as high in the subsequent two to three years”

Comment 9

Section “Difference-in-difference (DID) model estimation results”: check first sentence.

Respond to the comment 9

Many thanks for the reviewer’s careful comments. We revise the first sentence as following:

“Although SEPAP’s intervention covers 471 counties, there are missing data in several variables in the China County Statistical Yearbook, such as rural per capita disposable income of numerous counties, including the pilot counties in Qinghai and Tibet”.

Reviewer 2#

Comment 1

In general, the paper is well written. But the program needs to be described in details. Also, the results should be discussed with regard to the existing literature.

Respond to comment 1

Many thanks for the reviewer's constructive comments. We search a number of references and provide more details for the program:

“In 2015, the scale of China's SEPAP program reaches 1.84GW (Li, 2019). By the end of 2018, a total of 15.44 million kW of photovoltaic poverty alleviation has been allocated nationwide, among which 13.63 million kW of grid-connected photovoltaic poverty alleviation projects have been completed in 26 provinces, and 2.24 million poor households registered (Qin and Wu, 2019)”.

As for the results, we discuss with regard to the existing literature:

“The effect of targeted PV poverty alleviation on the natural logarithm of annual rural per capita disposable income is positive with a coefficient of 0.0724, which is statistically significant at the 1% level. The coefficient is decreased to 0.0446 when the interaction term between SEPAP variable and the exposure time is included. The results extend the findings of Geall (2016), Zhou and Liu (2018) , Xu et al. (2019), and Li (2019) by using quantitative analysis and highlighting the importance of government intervention. Our study is quite different from that of Liao and Fei (2019), who focus on the installation capacity instead of income of poverty-stricken families, although we reach the similar conclusion”.

“The sunlight exposure time is positively correlated with the annual rural per capita disposable income, consistent with the Bridge et al.(2016), which measure the natural conditions and income with annual global solar radiation and consumption per capita, respectively”.

Comment 2

The paper lacks a theoretical framework that should establish the theory of change underlining this evaluation. Using this theoretical framework, the paper can identify the channels through which the SEPAP could affect the disposal income. In this program, there are two ways that can be analysed: (i) the direct effect due to the income distribution from which households could benefit, or (ii) the indirect effect due to the income effect of benefiting from electrification.

For the first point, the effectiveness of the income distribution scheme depends on the system of governance established at village level; this issue should be accounted for. For the second issue, the author(s) can build on Diallo & Moussa (2020), Lenz et al. (2017), Peters & Sievert (2016) and Bridge et al. (2016). In these papers, the way access to electricity in general or to solar energy in particular in off grid might affect income are clearly described.

The lack of a theoretical framework led to a misspecification of the evaluation model. In fact, the author(s) state(s) that “This index combines three key factors: proportion of the population in a county in poverty, rural net income per capita...”. This implies that the selection of beneficiaries (counties or villages) is endogenous (since it depends on income). Then, the impact assessed is potentially biased.

Respond to comment 2

Many thanks for the reviewer’s constructive suggestions. We build a theoretical framework by drawing on the studies of Diallo and Moussa (2020), Lenz et al. (2017), Peters and Sievert (2016) and Bridge et al. (2016). The revised parts are shown in the section of research hypotheses.

“To identify the channels through which the SEPAP could affect the disposal income, we analyze the two effects: (i) the direct effect due to the income distribution from which households could benefit, or (ii) the indirect effect due to the income effect of benefiting from electrification. As for the SEPAP program established through village-level arrays, the power stations sell generated electricity to the state grid company in full amount, and the later on pay the purchase price which granted with feed-in tariff by the government to the village collectives. Then, as stated in the section of program details, the village collectives distribute most of earnings to the eligible poverty-stricken families in the forms of public welfare posts, small public welfare undertakings, small and micro awards directly. But, there is also an indirect effect (see Figure 2). The SEPAP program may entitle poverty-stricken families to better access to knowledge and information, which bring better income opportunities, new business, and income improvement to the beneficiaries.

Figure 2 Channels through which household income is affected by the SEPAP.

Source: Our own adaptation from Diallo and Moussa (2020), Lenz et al. (2017), Peters and Sievert (2016) and Bridge et al. (2016)”.

Comment 3

The issue of the exposure time (to the program) should be addressed since the program had been set up from 2013 to 2016. Including a year fixed effect is not sufficient. An interaction between SEPAP variable and the exposure time is more likely to control for this issue (see King & Jere, 2009). In such a situation, the impact of the program is time dependent; since (i) the village collective might improve their governance of the distribution system and (ii) the impact of access to electricity on income can be observed on long run (Van der Walle et al. 2017).

Respond to comment 3

Many thanks for the reviewer's insightful comments. We re-estimate the results by considering the interaction between SEPAP and the exposure time (to the program). The updated results are shown in Table 1. We also cite the studies of King and Jere (2009) and Van der Walle et al. (2017) in the revised manuscript.

The explanation for the effect of interaction between SEPAP and the exposure time is as following:

“The issue of the exposure time (to the program) should be addressed since the program had been set up from 2013 to 2016 and the counties start to be covered by the SEPAP in different times. Model (3) adds an interaction item between SEPAP and the exposure time DURATION following King and Jere (2009). In such a situation, the impact of the program is time dependent, since (i) the village collectives might improve their governance of the distribution system and (ii) the effect of access to electricity on income can be observed in the long run (Van de Walle et al., 2017). The interaction item between SEPAP and the exposure time is significant, with a coefficient of 0.0274”.

Comment 4

The choice of control variables needs to be motivated. In particular, the choice of the per capita GDP of the province. Also, the choice of the sunlight exposure time instead of the average solar radiation (that is the more determinant for the PV productivity), see for example Bridge et al. (2016).

Respond to comment 4

In the revised manuscript, we provide more explanations for the choice of control variables:

“The regional industrial structure is measured by the proportion of the value added of secondary industries in regional GDP. A high value indicates a great ability of secondary industries to absorb the employment of rural and urban population, which is conducive to the transfer of rural poor population to the city, reducing the poverty alleviation burden of PV-based poverty alleviation project. Therefore, this variable is expected to improve poverty alleviation effect”.

“The SEPAP program requires huge investment, and its cost is composed of photovoltaic equipment (including modules, inverters and PV supporting bracket, etc.), engineering procurement construction cost (including civil works, installation, design, access system and project construction management fee, etc.), land use right and capitalized interest, all of which need the funds support. Take one program of GCL new energy holdings limited as a case, the total investment is 179.79 million yuan, among which the equipment investment accounts about 83.39% of the total costs, followed by the engineering procurement construction, 11.69%. The expenditure of poverty alleviation funds is measured by the proportion of public financial expenditure and income. The larger this value is, the greater the amount of poverty alleviation funds that can be invested. Ceteris paribus, we expect this variable to have a positive impact on poverty alleviation”.

“Abundant solar resources in a region indicate high PV power generation ability. We expect this variable to have a positive effect on local household income. Both sunlight exposure and average solar radiation are the indicators measuring the abundance of natural conditions. Although the

average solar radiation is recognized as one of the determinants for the PV productivity, which has been used by Bridge et al. (2016), this indicator is unobservable directly and can only be obtained by converting from the sunlight exposure time. The conversion is complicated that should incorporate a number of meteorological parameters such as cloud and precipitation, etc. Therefore, we use sunlight exposure time as a proxy for natural conditions”.

“Regarding land investment, the area under cultivation (agriculture) can be used as an indicator to measure the investment status of PV poverty alleviation. Land under cultivation refers to land with production facilities for crop cultivation or aquaculture. PV poverty alleviation is feasible not only due to solar panels installed on roofs of farmers, barren mountains and deserts, but also on crop cultivation greenhouses or aquaculture fish ponds. **More land rent will contribute to large-scale power generation, for example, the village-level plants joint construction arrays will generate more electricity than that of rooftop projects.** In theory this indicator is positively correlated with rural per capita disposable income”.

“Through education, low-income individuals may acquire skills and knowledge that enhance the efficacy of industrial poverty alleviation policies (Ge and Lei, 2013). Education level indicators include number of middle school students, number of primary school students, middle school teachers, and primary school teachers. Compared to the number of primary and secondary school teachers, the proportion of the number of secondary school students in the total population can highlight the local educational output.”.

“In terms of regional macroeconomic conditions, **we use per capita GDP of the province where the county is located. Several indicators, such as economic growth rate, unemployment rate, and inflation rate reflect the regional macroeconomic conditions. In comparison, the per capita GDP of the provinces where the county is located takes regional population into account, and this would be better at highlighting the level of economic development. Better regional macroeconomic performance may have a positive spillover effect on poverty alleviation**”.

Comment 5

In Tables 1, 2, S5, S6, S7, S8, S9 and S10, values in parentheses are not standard errors (there are some negative values). Maybe the author(s) mean t-statistics or z-statistics?

Since the SEPAP variable is a binary variable and the main outcome is the log of disposable income, the coefficient β associated with the SEPAP variable should not be interpreted directly. The authors should interpret $100*(e^{\beta} - 1)$ instead; see for example Diallo & Moussa (2020) or Bridge et al. (2016).

Respond to comment 5

Many thanks for the reviewer’s careful work. The values in parentheses are t-statistics instead of standard errors and we have corrected. Since the SEPAP variable is a binary variable and the main outcome is the natural logarithm of disposable income, the coefficient β associated with the SEPAP variable should not be interpreted directly. In the revised paper, we use t-statistics to replace standard error and interpret $100*(e^{\beta} - 1)$ as following:

“Model (1) finds that the PV poverty alleviation policy is associated with an improved rural disposable income of approximately 7.52%. Model (2) adds a set of additional control variables, which include the degree of marketization across China’s regions (a measure consisting of non-state economy, the development of product market, the development of factor market, the development of market intermediary organizations, legal environment, and the relation between government and market (Wang et al., 2017), solar resource (measured as annual solar exposure), and per capita GDP of the province. Model (2) yields a similar policy effect estimate of 7.51%”.

“This model estimates that the intervention of PV poverty-alleviation policy increases rural disposable income by 7.51%, which is marginally higher than the estimates in Table 1”.

“In the three years after the implementation of SEPAP, the natural logarithm of per capita disposable income increases, from 9.13% increase in the year of implementation to 21.80% three years after implementation (the estimated effect is slightly higher in the second year after implementation, 23.78%)”.

“Empirical results reveal that the PV poverty alleviation policy has a greater effect on the poor regions, and the effect amounts to 4.86%”.

We hope the explanations could meet the editor and reviewers’ requirements. Thanks editors again for the giving us the opportunity to revise the manuscript. Also, we really appreciate the reviewers’ constructive comments. If you have any further questions, please do not hesitate to let us know.

Best

Yueming

Reviewers' comments:

Reviewer #1 (Remarks to the Author):

After considering the response to all the comments made by the authors, I believe that the paper is ready for publication in Nature Communications.

Reviewer #2 (Remarks to the Author):

The authors have addressed almost all the points I raised in my previous report. However, there is an important issue that should be addressed.

I stated in the previous report that "In fact, the authors state that "This index combines three key factors: proportion of the population in a county in poverty, rural net income per capita...". This implies that the selection of beneficiaries (counties or villages) is endogenous (since it depends on income). Then, the impact assessed is potentially biased."

This question implies that the County selection criteria should be clearly explained and the authors should provide evidence that the SEPAP variable is not endogenous. In case the SEPAP variable is really endogenous, the authors should implement an endogenous treatment effect model to account for the endogeneity bias.

Response to reviewers' comments

Reviewer #2 (Remarks to the Author):

Comment 1

The authors have addressed almost all the points I raised in my previous report. However, there is an important issue that should be addressed.

I stated in the previous report that “In fact, the authors state that “This index combines three key factors: proportion of the population in a county in poverty, rural net income per capita...”. This implies that the selection of beneficiaries (counties or villages) is endogenous (since it depends on income). Then, the impact assessed is potentially biased.”

This question implies that the County selection criteria should be clearly explained and the authors should provide evidence that the SEPAP variable is not endogenous. In case the SEPAP variable is really endogenous, the authors should implement an endogenous treatment effect model to account for the endogeneity bias.

Respond to comment 1

Many thanks for the reviewer's constructive suggestions concerning the endogeneity issues. We are very sorry that this question was not satisfactorily resolved in the first round. We have an in-depth discussion about this question and propose the following solutions.

(1)“The index combines three key factors: proportion of the population in a county in poverty, rural net income per capita...”. **This criterion refers to the selection of national poverty-stricken counties rather than the Solar Energy for Poverty Alleviation Program (SEPAP) counties. Therefore, we have deleted this sentence from the revised paper.** The selection criteria for the SEPAP poverty alleviation counties is stipulated in “Opinions of Photovoltaic Poverty Alleviation Work File”, jointly issued by the National Development and Reform Commission, the State Council Leadership Group for Poverty Alleviation and Development, the National Energy Administration, the China Development Bank, and the Agricultural Development Bank of China(see the section Program Details): “By 2020, specifically in the areas both with previous PV poverty alleviation pilot projects and better sunlight conditions, the program should boost overall-village incomes for about 35 thousand poverty-stricken villages (for which poverty files have been established) located in 471 counties in 16 provinces. Each of the 2 million poverty-stricken families without capacity to work and for which poverty files have been established (including the handicapped) shall earn an additional income of at least 3,000 yuan per household each year from the program”¹. **This implies that the sunlight condition is the first order**

¹ For more information on the policy “Opinions of Photovoltaic Poverty Alleviation Work File”, please see [website link: http://www.gov.cn/xinwen/2016-04/02/content_5060857.htm](http://www.gov.cn/xinwen/2016-04/02/content_5060857.htm)

determinant, and the local economic condition is the secondary consideration for the selection of SEPAP poverty alleviation counties. Moreover, as of 2016, there are 473 national-level poverty-stricken counties in total in China, among which 175 counties are pilot counties for PV poverty alleviation, accounting for less than 40% of total national-level poverty-stricken counties. This also proves that the selection of SEPAP poverty alleviation counties is not solely dependent on the economic condition.

Despite of this, the reviewer's suggestions are very insightful. We therefore implement an endogenous treatment effect model to account for the endogeneity bias, and the natural logarithm of sunlight hours and per capita GDP of the province where the county is located are included in the equation for the treatment effect. These two variables are highly correlated with the selection of the SEPAP in China. Table S11 presents the relevant results. We also added the following discussion in the Supplementary Information.

In Column (1), the estimated coefficient of SEPAP amounts to 0.1471, which is significant at the 1% level. The economic magnitude of the treatment effect is twice of that in the baseline results without accounting for the endogenous treatment effects, indicating potential underestimate of the effect of SEPAP on rural household income in the baseline models. The result of treatment effects equation shows that sunlight hours are significantly positively correlated with the likelihood of being selected into the SEPAP. Column (2) adds natural logarithm of provincial per capita GDP into the equation for treatment effects, and the coefficient of SEPAP increases to 0.1630, which is significant at the 1% level. The result of treatment effects equation shows that provincial per capita GDP is negatively correlated with the likelihood of being selected into the SEPAP. The result is also consistent with our conjecture since the SEPAP is partially determined by the local economic conditions.

Table S 11 Endogenous treatment effect

	(1)	(2)
	Ln(DISINRURAL)	Ln(DISINRURAL)
SEPAP	0.1471*** (3.09)	0.1630*** (2.99)
SECONDDGPR	0.3331*** (5.29)	0.3332*** (5.29)
PUBEXINR	-0.0167*** (-6.75)	-0.0167*** (-6.75)
LNAGACRE	0.0302*** (6.75)	0.0302*** (6.75)
EDUCATION	1.5658** (2.28)	1.5647** (2.28)
MKTINDEX	0.0701*** (8.77)	0.0701*** (8.77)
LNSUNHOUR	-0.0549* (-1.89)	-0.0551* (-1.89)
LNGDPPROVINCE	0.2101*** (5.87)	0.2106*** (5.89)
Equation for treatment effects		
SEPAP		
LNSUNHOUR	2.2861*** (2.79)	1.2863*** (3.22)
LNGDPPROVINCE		-1.6510*** (-2.63)

County FE	Yes	Yes
Year FE	Yes	Yes
Observations	3,211	3,211
Number of Counties	865	865
Log Likelihood	-0.24	5.42

Notes: The dependent variable is the natural logarithm of disposable income of rural people per capita. SEPAP represents whether or not a county was selected for the photovoltaic poverty alleviation policy in a specific year. SECONGDPR depicts a proportion of the added value of the secondary industry to GDP. PUBEXINR shows the ratio of public expenditure to revenue. LN(AGACRE) examines the land used for facility agriculture facility agriculture land. EDUCATION estimates the ratio of number of secondary school students to the total population. MKTINDEX represents marketization index. LN(SUNHOUR) indicates sunlight exposure time. LN(GDPPROVINCE) is used to investigate the per capita GDP of the province where the county is located. ***, **, and * represent the significance levels of 1%, 5% and 10%, respectively. T-statistics are reported in parentheses.

(2) Additionally, we checked for the parallel trend assumption in our baseline DID methods to ensure that the treatment and control groups had similar trends in the disposable income prior to the treatment. Further, we used propensity score matching to deal with any potential systemic differences between the treatment group and control group, and the DID estimation results for the matched samples remain quantitatively similar (see Table 2).

We hope the explanations could meet the editor and reviewers' requirements. We thank the editors and the second reviewer again for giving us the opportunity to revise the manuscript. Also, we really appreciate the reviewers' constructive comments. If you have any further questions, please do not hesitate to let us know.

Best
Yueming

REVIEWERS' COMMENTS:

Reviewer #2 (Remarks to the Author):

Thank you for these clarifications.
I have no more comment on the paper.

Reviewer #2 (Remarks to the Author):

Thank you for these clarifications.
I have no more comment on the paper.

Response: Thank you for helping improve our manuscript.